# Effectiveness Evaluation of a UV-C-Photoinactivator against Selected ESKAPE-E Pathogens

**DOI:** 10.3390/ijerph192416559

**Published:** 2022-12-09

**Authors:** Karyne Rangel, Fellipe O. Cabral, Guilherme C. Lechuga, Maria H. S. Villas-Bôas, Victor Midlej, Salvatore G. De-Simone

**Affiliations:** 1Center for Technological Development in Health (CDTS)/National Institute of Science and Technology for Innovation in Neglected Population Diseases (INCT-IDPN), Oswaldo Cruz Foundation (FIOCRUZ), Rio de Janeiro 21040-900, RJ, Brazil; 2Laboratory of Epidemiology and Molecular Systematics (LESM), Oswaldo Cruz Institute, Oswaldo Cruz Foundation (FIOCRUZ), Rio de Janeiro 21040-900, RJ, Brazil; 3Health Sciences Center, Institute of Microbiology Paulo de Góes, Federal University of Rio de Janeiro (UFRJ), Rio de Janeiro 21941-853, RJ, Brazil; 4Microbiology Department, National Institute for Quality Control in Health (INCQS), Oswaldo Cruz Foundation (FIOCRUZ), Rio de Janeiro 21040-900, RJ, Brazil; 5Structural Biology Laboratory (LBE), Oswaldo Cruz Institute, Oswaldo Cruz Foundation (FIOCRUZ), Rio de Janeiro 21040-900, RJ, Brazil; 6Post-Graduation Program in Science and Biotechnology, Department of Molecular and Cellular Biology, Biology Institute, Federal Fluminense University (UFF), Niterói 22040-036, RJ, Brazil

**Keywords:** UV-C, shoe sole decontaminator, ESKAPE-E pathogens, multidrug resistance, disinfection, cell viability, SEM

## Abstract

Healthcare-associated infections (HAI) worldwide includes infections by ESKAPE-E pathogens. Environmental surfaces and fomites are important components in HAI transmission dynamics, and shoe soles are vectors of HAI. Ultraviolet (UV) disinfection is an effective method to inactivate pathogenic microorganisms. In this study, we investigated whether the SANITECH UV-C shoe sole decontaminator equipment that provides germicidal UV-C radiation could effectively reduce this risk of different pathogens. Six standard strains and four clinical MDR strains in liquid and solid medium were exposed to a UV-C System at specific concentrations at other times. Bacterial inactivation (growth and cultivability) was investigated using colony counts and resazurin as metabolic indicators. SEM was performed to assess the membrane damage. Statistically significant reduction in cell viability for all ATCCs strains occurred after 10 s of exposure to the UV-C system, except for *S. enterica*, which only occurred at 20 s. The cell viability of *P. aeruginosa* (90.9%), *E. faecalis* and *A. baumannii* (85.3%), *S. enterica* (82.9%), *E. coli* (79.2%) and *S. aureus* (71.9%) was reduced considerably at 20 s. In colony count, after 12 s of UV-C exposure, all ATCC strains showed a 100% reduction in CFU counts, except for *A. baumannii*, which reduced by 97.7%. A substantial reduction of colonies above 3 log_10_ was observed at 12 and 20 s in all bacterial strains tested, except for *A. baumannii* ATCC 19606 (12 s). The exposure of ATCCs bacterial strains to the UV-C system for only 2 s was able to reduce 100% in the colony forming units (CFU) count in all ATCCs strains, *S. aureus*, *P. aeruginosa*, *E. coli*, *A. baumannii*, *E. faecalis*, except the *S. enterica* strain which had a statistically significant reduction of 99.7%. In ATCC strains, there was a substantial decrease in colonies after 4 s (sec) of exposure to the UV-C system, with a reduction ranging from 3.78–4.15 log_10_ CFU/mL. This reduction was observed in MDR/ESKAPE-E strains within 10 s, showing that UV-C could eliminate above 3.84 log_10_ CFU/mL. SEM showed a reduction of pili-like appendages after UV-C treatment in all strains except for *E. coli* (ATCC 25922). The Sanitech UV-C shoe sole decontaminator equipment from Astech Serv. and Fabrication Ltd. (Petrópolis, Brazil), effectively killed in vitro a series of ATCCs and MDR/ESKAPE-E bacteria of sanitary interest, commonly found in the hospital environment.

## 1. Introduction

The leading cause of Healthcare-associated infections (HAI) worldwide includes infections by Enterococcus faecium, Staphylococcus aureus, Klebsiella pneumoniae, Acinetobacter baumannii, Pseudomonas aeruginosa, Enterobacter spp. and Escherichia coli (ESKAPE-E) [1]. ESKAPE-E pathogens embody the top five bacterial species and genus with relevant intrinsic resistance and expansive ability to acquire multidrug resistance—prioritized as a global health threat in urgent need of new antibiotic research and development by the World Health Organization (WHO) [2].

Environmental surfaces and fomites are important components in HAI transmission dynamics [3]. One such fomite is the shoes worn by healthcare professionals, patients and visitors. Shoe soles are possible vectors of infectious diseases containing a high biological load of microorganisms [4,5,6]. Agarwal et al. performed bacterial isolation and quantification of operating room boots and found that most surgical boots were contaminated with normal human microflora, including *Staphylococcus*, *Streptococcus*, and *Bacillus* species [5]. Amirfeyz et al. [7] analyzed bacterial contamination of operating room shoes at the beginning and end of the workday compared to outdoor shoes. They reported that 88% of outdoor shoes were positive for at least two pathogenic bacteria. About 48% of operating room shoes were also positive for at least one pathogenic species, most commonly coagulase-negative *Staphylococcus* [7]. A study verified contamination of doctors’ shoe soles before and after medical patient care visits using swab samples processed by Poland’s National Coordination Center, identifying methicillin-resistant *Staphylococcus aureus* (MRSA) and *Enterococcus faecalis* on the soles of shoes on 56% of physicians before and 65% after doctor’s rounds [8]. In another study of 41 doctors and nurses in an intensive care hospital, shoe soles were positive for at least one pathogen in 12 (29.3%) participants; MRSA was the most common.

Furthermore, 98% (49/50) of shoes worn outdoors showed positive bacterial cultures compared to 56% (28/50) of shoes reserved for the operating room only [9]. Studies carried out as early as the 1970s demonstrated that the redistribution of airborne bacteria from the functional room floor accounted for up to 15% of all airborne bacteria [10]. Walking on contaminated floors was a more effective aerial dispersal than mopping or sweeping. When examining the effects of ventilation on airborne pathogens in the operating room, it was found that 15% of the airborne bacteria in the operating room originated from operating room floors [11].

Hair and shoes can act as vehicles for transmitting pathogens [12] and sharing the SARS-CoV-2 that causes COVID-19, but recommendations for hair coverings and shoes to prevent SARS-CoV-2 are lacking [13]. In a study that evaluated SARS-CoV-2 on hospital floors, 70% (7/10) of intensive care unit (ICU) floor samples were positive in quantitative polymerase chain reaction (PCR) assays. Furthermore, 100% (3/3) of swabs taken from hospital pharmacy floors without COVID-19 patients were positive for SARS-CoV-2, meaning that contaminated shoes likely served as vectors [14]. In addition, SARS-CoV-2 nucleic acid was frequently detected on floors and high-touch surfaces inside COVID-19 rooms and on feet and shoes outside patient rooms in COVID-19 units [15].

Multidrug-resistant organisms (MDR), including *Clostridium difficile*, are on the soles of the shoes of healthcare workers and people living in the community [4]. Studies in healthcare settings and non-health community settings have demonstrated the presence of MRSA, *Vancomycin-Resistant Enterococci* (VRE), and Gram-negative MDR bacteria on shoe soles [8,16]. Toxigenic strains of *C. difficile* have been found in some shoe samples from non-healthcare homes in Houston, Texas, USA [6,17,18,19]. Buchler et al. demonstrated contamination in 17.8% of the soles of healthcare workers’ shoes with toxigenic strains of *C. difficile* linked epidemiologically and confirmed by whole genome sequencing in infected patients [20]. A high level of shoe sole contamination with *C. difficile* has also been observed in hospitals in Slovenia [21,22]. Several studies have also investigated the dynamics of transmission between shoe soles or floor surfaces and patient colonization, with most studies demonstrating the potential for colonization by aerosolization, direct contact, or indirect methods [4]. Others have shown that shoe soles are a source of contamination of floor surfaces and environmental contamination of soil in hospital and non-hospital settings [7,23,24,25,26]. The study by Chambers et al. revealed that microbiological pathogens on shoe soles can be transferred to a linoleum floor [27]. From the floor, it is plausible that drafts, human movements on the floor, and other factors that aerosolize or provide an aerial opportunity for the organism can occur, causing human infections by inhalation, horizontal or cross-contamination of other people, clothing, or equipment where the organism settles [27]. Furthermore, it is possible that due to the existence of these microbiological pathogens on the soles of shoes, the rapid spread of these organisms in the healthcare environment could be directly related to the fact that organisms on floors are picked up and transported by the soles of shoes and retransferred to bases in other areas by human movement [27]. Shoes become contaminated by a dirty floor, and parallel methods to decontaminate the bottom and new measures to prevent such spread are also needed.

High-level disinfection of shoes is difficult and, in most cases, incompatible with the composition of the shoes [4], mainly due to the relative lack of consistent efficacy in decontaminating shoe soles using chemical or non-chemical strategies [27]. Sticky rugs for operating rooms have been discouraged [28], and disposable shoe protectors can lead to even greater contamination of the environment and hands [29]. Shoe covers are removed by healthcare workers’ hands, leading to contamination of their hands that were likely free of pathogens [29]. Certainly, an effective disinfection strategy for shoe soles is urgently needed. Therefore, an innovative future approach could be using an ultraviolet C (UV-C) device to decrease the contamination of shoe soles [30].

Ultraviolet (UV) disinfection is recognized by the United States Environmental Protection Agency (US EPA) as a proven technology to decrease the risks of waterborne illness from microbial pathogens and the risk of exposure to disinfection by-products. Furthermore, it is an effective method to inactivate pathogenic microorganisms [31,32]. The UV spectrum is traditionally divided into four bands: UV-A (315–400 nm), UV-B (280–315 nm), UV-C (200–280 nm), and vacuum UV (VU-V, 100–200 nm) [33,34]. UV-C irradiation has been widely used as a primary disinfectant in drinking water and wastewater treatment to achieve effective inactivation of various pathogenic microorganisms, including bacteria, viruses, and protozoa [35]. The most effective germicidal wavelengths depend on the species of microorganisms and range predominantly between 260 and 280 nm [36]. A significant amount of research has been carried out in this range, as protein has a main peak at 280 nm while DNA peaks in the UV absorption curve at 260 nm [36]. Several studies have used higher wavelengths (280 nm) where generating UV irradiation is cheaper compared to lower UV-LED wavelengths (254–280 nm). However, in order to reduce the harmful effects of mercury, to comply with the United Nations Environment Programme (UNEP) on forbidding mercury products and new UV-emitting sources (e.g., UV light-emitting diode (UV-LED), pulsed xenon-based ultraviolet light (PX-UVC)) was introduced as a replacement of the conventional UV-C lamp [37].

UV-C devices are used in hospitals to decontaminate the hospital environment [38,39]. However, only some effective strategies are available to decontaminate shoe soles. Based on what has been described and given the germicidal potential of this technology on surfaces, in liquids, and in the air, and the possibility that shoe soles are vectors of HAI, in this study, we investigated whether the SANITECH UV-C shoe sole decontaminator equipment that provides germicidal UV-C radiation was able to reduce this risk effectively in different pathogens.

## 2. Materials and Methods

### 2.1. Bacterial Strains

Bacterial strains (Staphylococcus aureus (ATCC 6538), Pseudomonas aeruginosa (ATCC 15442), Salmonella enterica subsp. enterica serovar Choleraesuis (ATCC 10708), Escherichia coli (ATCC 25922), Acinetobacter baumannii (ATCC 19606), and Enterococcus faecalis (ATCC 29212) were obtained from the American Type Culture Collection (ATCC) (Plast Labor Ind. Com. EH Lab. Ltd.a, Rio de Janeiro, Brazil). In addition, representative MDR strains of the ESKAPE-E group were also used, with three clinical strains isolated from HAIs, which were: methicillin-resistant *S. aureus* (MRSA), carbapenemase-producing *K. pneumoniae* (KPC+), A. baumannii pandrug-resistant (PDR- nonsusceptibility to all agents in all antimicrobial categories), carrying the bla_OXA-23_ gene and representing one of the genotypes disseminated in Brazil (ST15/CC15), and an environmental strain of *P. aeruginosa* extremely drug-resistant (XDR- nonsusceptibility to at least one agent in all but two or fewer antimicrobial categories) from hospital effluent. These strains were kindly provided by Dr. Maria H. S. Villas Bôas (National Institute for Quality Control in Health of the Oswaldo Cruz Foundation—INCQS/FIOCRUZ) and Dr. Catia Chaia de Miranda (Interdisciplinary Medical Research Laboratory, LIPMED, FIOCRUZ). These bacterial strains were initially cultivated according to the instructions of the ATCC, aliquoted, and stored in cryotubes containing tryptic soy broth (TSB, Difco) with 20% glycerol (*v*/*v*) and kept at −20 °C for later use.

### 2.2. UV-C System

For the experiment, we used UV-C equipment (Sanitech UV-C shoe sole decontaminator, Astech Serv and Fabrication Ltd., Petrópolis, Brazil) with external dimensions of 45 cm × 47 cm × 13 cm containing six OSRAM UV-C lamps (three on each side) (HNS 8W G5 G5/PURITEC HNS UV-C|UV-C lamps for purification) (Appendix A) and dominant wavelength of 254 nm (OFR version). In this study, the equipment was used in an inverted position, with the upper platform facing downwards (UV-C System). The total UV-C intensity of the equipment was 15.150 µW/cm^2^.

### 2.3. Exposure to the UV-C System in a Liquid Medium

The strains were removed from the freezer stock culture for bacterial reactivation, sown in Triptone Soy Agar (TSA; DIFCO Laboratories Inc., Detroit, MI, USA), and incubated at 37 °C for 24 h. After the microorganisms were suspended in sterile Phosphate Buffered Saline (PBS), pH 7.0, the concentration of 10^8^ colony-forming units (CFU)/mL was determined with a densitometer (Densichek Plus, BioMérieux, Rio de Janeiro, Brazil). Subsequently, the suspensions were diluted in PBS on a scale of 1:100 to obtain a final cell concentration of 10^6^ CFU/mL, respectively, for each strain. Fifty-microliter aliquots of each ATCC bacterial suspension (*S. aureus*, *P. aeruginosa*, *S. enterica*, *E. coli*, *A. baumannii*, and *E. faecalis*) were inoculated into the microplate (96-well) in triplicate. The chosen UV-C exposure times took into account the minimum and maximum time that would be reasonable for the individual to wait on the shoe sole decontamination equipment. The time maximum time of the decontaminator timer was also tested. Each microplate was exposed to the UV-C system at times of 2, 4, 6, 8, 10, 12, and 20 s and incubated at 37 °C for 24 h. As a positive control of the assay, microplates containing the same bacterial suspensions were used without exposure to the UV-C system. Wells containing only PBS were used as negative controls. All microplates were exposed to UV-C separately, always following the same location of exposure to the UV-C system (left side, centered below the 3 lamps, numbering turned to the left and letters down), as well as the distribution of strains on the microplate (Appendix A). The distance between the UV-C lamps and the microplate was 3 cm. After exposure or not to the UV-C system, the microplates were incubated at 37 °C for 24 h. The assays were repeated three times.

#### 2.3.1. Cell Viability

After exposure of microplates containing bacterial strains to the UV-C system, cultures were diluted in LB broth (1:10, 1:100, and 1:1000) in triplicate. Then, the microplates were incubated at 37 °C for 24 h. As a positive control of the assay, we performed the same procedure with the microplates that were not exposed to the UV-C system. After incubation, bacterial growth was detected by adding 0.02% resazurin (7-hydroxyphenoxazin-3-one 10-oxide; Sigma-Merck, St. Louis, MO, USA) with 30 min at 37 °C [40]. Again, we used LB broth and PBS as a negative control, and the measurement at 590 nm was conducted on an ELISA plate reader (Flex Station 3; Molecular Devices, San José, CA, USA).

#### 2.3.2. Quantification of Colonies

Five microliter aliquots of each ATCC bacterial suspension (*S. aureus*, *P. aeruginosa*, *S. enterica*, *E. coli*, *A. baumannii*, and *E. faecalis*) were obtained in the previous test, item 2.3.1. (treated or not with UV-C), at different dilutions (1:10, 1:100, and 1:1000 in LB broth), were inoculated into Petri dishes containing TSA. The plates were incubated at 37 °C for 24 h for CFU.

### 2.4. Exposure to the UV-C System in Solid Medium

The strains were removed from the freezer stock culture for bacterial reactivation, sown in TSA, and incubated at 37 °C for 24 h. After the microorganisms were suspended in sterile 0.85% saline, the concentration of 10^8^ CFU/mL was determined with a densitometer (Densichek Plus, BioMérieux, Rio de Janeiro, Brazil). In LB broth, successive dilutions (10^4^ and 10^3^ CFU/mL) were made. Ten microliter aliquots of each bacterial suspension (*S. aureus*, *P. aeruginosa*, *S. enterica*, *E. coli*, *A. baumannii*, *E. faecalis, S. aureus* (MRSA), *P. aeruginosa* (XDR), *A. baumannii* (PDR) and *K. pneumoniae* (KPC+)) were plated in triplicate on TSA. Each petri dish was exposed to the UV-C system at times of 2, 4, 6, 8, and 10 s at a distance of 3 cm from the UV-C lamps. The plates were incubated at 37 °C for 24 h for CFU. As a positive control of the assay, TSA plates containing the same bacterial suspensions were used but without exposure to the UV-C system. In these control plates, the inoculum was plated by a spread plate. A plate containing only TSA was used as a negative control. The microbiological control of LB broth was also carried out.

### 2.5. Statistical Analysis

Each experiment was repeated three times for each bacterial strain in each treatment with the UV-C system. The collected data were analyzed using the program R (version 3.6.0) (Vienna, Austria) and R Studio, where the paired t-test was applied to compare the statistical significance between the two samples (with and without treatment with UV-C system) with ≤0.05. Percent reduction and Log_10_ CFU reductions from colonies were calculated as follows:$$\mathrm{Percentage}\;\mathrm{reduction}=\frac{\left(B-A\right)}{B}\;\times \;100$$
Reduction Log_10_ CFU = Log_10_ (B − A) CFU
where: B = Number of viable microorganisms. A = Number of viable microorganisms after UV-C irradiation.

### 2.6. Scanning Electron Microscopy (SEM)

Morphological changes in the bacteria species were visualized using SEM [41]. For analysis, control cells or UV-C system treatment (10^6^ CFU/mL at 2, 6, and 12 s of exposure) were fixed for 1 h with 2.5% glutaraldehyde (Sigma-Aldrich, Barueri, SP, Brazil) in 0.1 M cacodylate buffer (Sigma-Aldrich, Brazil). After fixation, the cells were washed three times in PBS for 5 min, post-fixed for 15 min in 1% osmium tetroxide (OsO4) (Electron Microscopy Sciences, Pennsylvania, PA, USA), and rewashed three times in PBS for 5 min. Next, the samples were dehydrated in an ascending series of ethanol (Merk, Brazil) (7.5, 15, 30, 50, 70, 90, and 99.9% ethanol) for 15 min each step, critical point dried with CO_2_, sputter-coated with a 15-nm thick layer of gold and examined in a Jeol JSM 6390 (Tokyo, Japan) scanning electron microscope.

## 3. Results

### 3.1. Exposure to the UV-C System in a Liquid Medium

#### 3.1.1. Cell Viability

Cell viability analysis in solution was evaluated by the spectrophotometric reading of the measurement of resazurin to resorufin reduction. In this assay, after UV-C system treatment, successive dilutions of cultures were made in LB broth, and after 24 h, the viability of the different species was measured. As it was not possible to count bacterial colonies in the 1:10 dilution and there was no significant difference between the 1:100 and 1:1000 dilutions, we selected for viability analysis the final dilution of bacterial cells of 1:100, respectively, for each bacterial strain (ATCCs). When taking into account the average of the readings performed at each study time (2 to 20 s), we verified that the statistically significant reduction in cell viability for all ATCCs strains occurred after 10 s of exposure to the UV-C system, except for *S. enterica* which only occurred at 20 s (Table 1). Therefore, this table presented only the results with statistics. At 10 s, we observed a reduction of 60.1% in *A. baumannii*, 50.2% in *S. aureus*, 44.9% in *E. faecalis*, 42.9% in *P. aeruginosa*, and 41.2% in *E. coli*. At 12 s, there was a greater reduction in viability compared to the last time, 50.5% for *E. coli*, 83.2% for *E. faecalis*, and 93.4% for *P aeruginosa*. However, at 12 s there was no statistically significant reduction for *S. aureus* and *A. baumannii* (Figure 1, Table 1). Treatment with 20 s of UV-C clearly significantly reduced bacterial growth in all strains studied, leading to an inhibition of about 90.9% in *P. aeruginosa*, followed by 85.3% in *E. faecalis*, and *A. baumannii*, 82.9% in *S. enterica*, 79.2% *in E. coli* and 71.9% in *S. aureus* (Figure 2, Table 1).

#### 3.1.2. Quantification of Colonies

The culture exposure (diluted 1:100) at different times (2 to 20 s) to the UV-C system was able to inhibit the in vitro growth of all bacterial strains tested (Figure 2) with a statistically significant reduction in colony count compared to the control group (not treated with UV-C system) (Table 2). Among the ATCCs strains, *P. aeruginosa* and *S. enterica* have already significantly reduced CFU/mL counts with just 2 s of UV-C exposure, with a reduction of 67.1% and 63.6%, respectively. The others did not suffer a significant decrease in the CFU/mL count. After 4 s of exposure to UV-C, there was already a reduction of at least 50% in the CFU/mL count of all ATCCs strains compared to the control group (not treated with the UV-C system). In the UV-C exposure time of 6 s, the greatest reductions were observed for *S. aureus* (91.9%) and *P. aeruginosa* (98.4%). At 8 s of UV-C exposure, the greatest reduction in CFU/mL counts was observed only for *S. enterica*, 90.6%, and *S. aureus*, 96.3%. However, the others showed lower reduction values than in the more down exposure time (6 s). At 10 s of UV-C exposure, all ATCC strains, except *E. faecalis* (76.9%), showed a significant reduction in the number of CFU/mL counts. After 12 s of UV-C exposure, all ATCC strains showed a 100% reduction in CFU/mL counts, except for *A. baumannii*, which reduced by 97.7%. At 20 s, there was no further colony growth (Figure 3, Table 2). The total reduction in log_10_ CFU/mL is illustrated in Figure 3. With 2 to 10 s of exposure, there was a reduction of 0.14–2.23 log_10_ CFU/mL. A substantial decrease in colonies above 3 log_10_ was observed at 12 and 20 s in all bacterial strains tested, except for *A. baumannii* ATCC 19606 (12 s) (Figure 2).

### 3.2. Exposure to the UV-C System in Solid Medium

The culture exposure at different times (2 to 20 s) to the UV-C system was able to inhibit the in vitro growth of all bacterial strains tested, ATCCs, and MDR/ESKAPE-E group (10^4^ CFU/mL) (Figure 4) with a statistically significant reduction in colony count compared to the control group (not treated with UV-C system) (Table 3). The exposure of ATCCs bacterial strains to the UV-C system for only 2 s was able to reduce 100% in the CFU/mL count in all ATCCs strains (10^4^ CFU/mL), namely, *S. aureus*, *P. aeruginosa*, *E. coli*, *A. baumannii*, *E. faecalis*, except the *S. enterica* strain which had a statistically significant reduction of 99.7%. At the exposure time of 4 s, this strain had its CFU/mL count reduced to 100%. At UV-C exposure times of 6, 8, and 10 s, the total reduction (100%) result remained constant (Figure 4, Table 3).

A different behavior was observed in the MDR strains representative of the ESKAPE-E group (10^4^ CFU/mL). The *S. aureus* (MRSA) and *K. pneumoniae* (KPC+) strains, when exposed to UV-C for 2 and 4 s, suffered a 100% reduction in the number of CFUs. However, at 6 s, they reduced by 99.8% and 99.5%, respectively. In times of 8 and 10 s, the result remained constant with a reduction of 100%. The same occurred with the strain *P. aeruginosa* (XDR), which significantly reduced from 2 s of exposure to UV-C, being 99.1%, followed by 99.3% (4 s), 99.1% (6 s), 99, 5% (8 s) and 100% (10 s). The *A. baumannii* (PDR) strain initially showed a 100% reduction in a UV-C exposure time of 2 s, but in the following times of 4 and 6 s, it reduced the CFU/mL count by 99.1% and 99.7%, respectively. At 10 s, we observed a 100% reduction in CFU/mL (Figure 4, Table 3). The total reduction in log_10_ CFU/mL is illustrated in Figure 5. In ATCC strains, there was a substantial decrease in colonies after 4 s of exposure to the UV-C system, with a reduction ranging from 3.78–4.15 log_10_ CFU/mL. This reduction was observed in MDR/ESKAPE-E strains within 10 s, showing that UV-C could eliminate above 3.84 log_10_ CFU/mL (Figure 6).

### 3.3. Scanning Electron Microscopy (SEM)

SEM was performed to confirm membrane damage to bacterial species. The strains of *S. aureus* (ATCC 6538), *P. aeruginosa* (ATCC 15442), *A baumannii* (ATCC 19606), *E. faecalis* (ATCC 29212), *S. aureus* (MRSA), and *P. aeruginosa* (XDR, after treatment with UV-C for 2 s, showed no precipitate (“pellet”) at the bottom of the microtube, being observed only in controls (untreated) and therefore could not be analyzed by SEM. The same occurred with all strains treated with 6 and 12 s of UV-C that did not produce pellets and, therefore, could not be analyzed under the microscope. These exposure times probably eliminated all strains, which agrees with the results found in the counts. Morphological analysis showed that *S. enterica* (ATCC 10708), *E. coli* (ATCC 25922), *A. baumannii* (PDR), and *K. pneumoniae* (KPC+) controls showed some pili-like appendages that vary in size and unevenly distributed. After UV-C treatment (2 s), all the strains presented membrane alterations with a reduction of the pili-like appendages, except for *E. coli* (ATCC 25922), which did not have its morphology altered (Figure 6 and Figure 7).

## 4. Discussion

In the face of HAIs, ultraviolet (UV) light has been widely explored as a practical innovation for disinfecting bacteria [42,43,44]. Several UV-C devices have been tested to decontaminate the hospital environment, including UV-C devices with UV-reflecting ink [45] or vector-specific UV-C such as stethoscopes [46,47]. Studies have shown that shoe soles can transmit pathogens to patients through direct or indirect transmission routes. However, only some effective strategies are available to decontaminate shoe soles. In the present study, the Sanitech UV-C shoe sole decontaminator was tested for its ability to disinfect different pathogens, demonstrating that it was effective in killing ATCCs bacterial strains (*S. aureus* ATCC 6538, *P. aeruginosa* ATCC 15442, *S. enterica* ATCC 10708, *E. coli* ATCC 25922, *A. baumannii* ATCC 19606, *E. faecalis* ATCC 29212) and IRAS-associated ESKAPE-E multidrug (*S. aureus* (MRSA), *P. aeruginosa* (XDR), *A. baumannii* (PDR)), *K. pneumoniae* (KPC+). A similar study that used a UV-C device specifically designed to decontaminate shoe soles was tested for its ability to decrease the number of CFUs in standardized rubber-soled shoe soles and subsequent transmission to floors or a simulated hospital room [30]. In this study, relevant pathogens, standardized procedures for inoculation of shoe soles, a variety of types of floors, and a standardized script to mimic hospital conditions were used. The results showed that experiments performed with shoes exposed to the UV-C device had CFU/mL significantly lower in the soles of shoes, floors, and patient care areas [30]. These results demonstrated that a UV-C device aimed at the soles of shoes can decrease subsequent environmental bioburden and patient colonization.

Several studies have demonstrated the inactivation of different microorganisms using UV-LEDs. The effect of UV stress on bacteria can be easily evaluated with UV-LED systems and a microplate that allows the simultaneous irradiation of multiple replicas at different UV doses (demonstrated in *E. coli* and MS2) [48], even though most studies use a default setup that allows only one sample to be irradiated at a time (i.e., plate or beaker), resulting in limited statistical power [49,50]. Therefore, for the analysis of cell viability, we used microplates (96 wells) where different bacterial strains could be exposed to the same conditions. Cell viability assay or cytotoxicity assay is a test that analyzes metabolically active cells in cell culture to assess their metabolic activity qualitatively and quantitatively.

According to our results, when we investigated the metabolic capacity of bacterial strains through resazurin, we observed that exposure to UV-C for 10 s was able to interfere with the cell viability of all ATCCs songs with a statistically significant reduction of metabolically active cells in cultures, except for *S. enterica* (ATCC 10708) (20 s). The exposure time of 20 s was the one that had the greatest interference in the cell viability of all ATCCs strains, with reduction rates ranging from 71.9% to 90.9%. Environmental stress is an external factor that hurts the physiological well-being of bacterial cells, leading to reduced growth rate or, in more extreme circumstances, inhibition and/or cell death at individual or population levels [51]. The exposure of bacterial cells to bactericidal UV radiation can be considered environmental stress, as this property has been used as a physical disinfection technique to eliminate entire bacterial populations, mainly from the clinical environment. Furthermore, various ecological stresses induce the ‘‘mar’’ (multiple antibiotic resistance) operons [52], which regulates the expression of several genes, including those that encode an efflux pump of broad specificity [53]. Therefore, bacterial cells have several mechanisms to select mutants within sublethally stressed bacterial populations, as well as to minimize stress and maximize continued cell viability to ensure survival after the stress conditions are removed [54]. This fact may explain what happened with the strains of *S. aureus* (ATCC 6538) and *A. baumannii* (ATCC 19606), which, despite showing a statistically significant reduction at 10 and 20 s, did not present at 12 s. Exposure of pathogenic bacteria to non-lethal stressful conditions can also increase resistance to the applied stress conditions [55,56].

Previous experimental studies showed that the duration of exposure to irradiation was a crucial determinant in the performance of UV-C equipment [57,58,59,60]. Furthermore, they also revealed that the shorter the distance from the agar plate to the UV-C equipment, the greater the bacterial killing efficiency the equipment could achieve [57,58]. In our study, the CFU count was derived from bacterial strains irradiated simultaneously on the microplate at different times (2, 4, 6, 8, 10, 12, and 20 s) but always at the same distance of 3 cm from the lamps. It is interesting to note that despite the low interference in the cell viability of bacterial strains in times of exposure to UV-C below 20 s, a short time of exposure to UV-C was able to interfere with the bacterial growth of cells, showing that in just 2 s UV-C already inhibited colony growth by 67.1% for *P. aeruginosa* (ATCC 15442), 63.6% for *S. enterica* (ATCC 10708). The other strains showed lower inhibition rates, as observed for *E. faecalis* (ATCC 29212) (39.7%), *S. aureus* (ATCC 6538) (30.3%), *A. baumannii* (ATCC 19606) (27.3%) and *E. coli* (ATCC 25922) (26.5%), however, at 4 s of UV-C exposure, their bacterial growth inhibition rates increased to 69.9%, 73.4%, 52.7 %, and 85%, respectively. *S. aureus* (ATCC 6538) had its bacterial growth inhibition rate raised to 73.4% and *P. aeruginosa* (ATCC 15442) to 91.8%. The highest reduction rates presented after 6 s of exposure were 98.4% for *P. aeruginosa* (ATCC 15442) and 91.9% for *S. aureus* (ATCC 6538). The other strains had an increasing reduction in CFU, ranging from 71.9% (*A. baumannii* (ATCC 19606) to 87.1% (*E. coli* ATCC 25922). After 8 s of exposure to UV-C, *S. aureus* (ATCC 6538) and *strains of S. enterica* (ATCC 10708) showed a greater reduction in CFU (96.3% and 90.6%) when compared to the last time (6 s). However, for the other strains of *P. aeruginosa* (ATCC 15442), *E. coli* (ATCC 25922), *A. baumannii* (ATCC 19606), and *E. faecalis* (ATCC 29212), the opposite occurred since they showed slightly lower rates of reduction. Within 10 s of exposure, there was a progressive increase in the rate of reduction of *S. enterica* ATCC 10708 (90.6% to 97.6%), followed by *S. aureus* (ATCC 6538) (96.3% to 97.7%) and *P. aeruginosa* (ATCC 15442) (96.6% to 99.4%). The reduction rates of *E. coli* (ATCC 25922) and *A. baumannii* (ATCC 19606) returned to increase above the percentage obtained at 6 s, except for *E. faecalis* (ATCC 29212). With 12 s of UV-C exposure, a 100% reduction was observed for all ATCCs strains tested (3 log_10_ CFU/mL), except *A. baumannii* ATCC 19606, which reduced 97.7% (1.65 log_10_ CFU/mL). In the longest exposure time used in this study (20 s), all ATCCs strains showed a 100% reduction in CFU. The results found in the CFU counting methodology showed different bacterial inactivation profiles to those observed with the cell viability method using resazurin for all ATCCs strains. Our results differ from those found by Toté et al. [61]. They developed a microplate resazurin method for evaluating the antimicrobial activity of antiseptics and disinfectants with different bacteria and reported that this method was as accurate as the plate counting method, presenting similar detection limits [61]. However, it is worth mentioning that the two tests are related to different measurement modes. While the CFU counting method considers bacterial growth, resazurin measures the number of cells through reactions associated with the oxidation of this compound in the intracellular medium. This fundamental difference between the measurement methods may cause not show the correspondence of the results.

In the CFU count of the solid medium assay, only 2 s of exposure to ATCCs bacterial strains were sufficient to eliminate all colonies of *S. aureus* (ATCC 6538) (4.15 log_10_ CFU/mL), *P. aeruginosa* (ATCC 15442) (3.96 log_10_ CFU/mL), *E. coli* (ATCC 25922) (3.88 log_10_ CFU/mL), *A. baumannii* (ATCC 19606) (3.83 log_10_ CFU/mL) and *E. faecalis* (ATCC 29212) (3.78 log_10_ CFU/mL), except for *S. enterica* (ATCC 10708) which suffered a reduction of 99.7% (2.46 log_10_ CFU/mL), being 100% (3.98 log_10_ CFU/mL) reduction achieved in 4 s. As in our study and despite the different conditions used, the study by Rashid et al. found that the soles of shoes exposed to a UV-C decontamination device showed a reduction in CFU counts of relevant pathogenic organisms, with a greater log_10_ reduction for *E. coli* (2.81 ± 0.80), followed by *S. aureus* (2.67 ± 0.81), *E. faecalis* (2.10 ± 0.62) and *C. difficile* (0.42 ± 0.68) [30]. We noticed that irradiation was much more effective in the solid medium, eliminating most of the CFU in ATCC bacterial strains in 2 and 4 s, while in the liquid medium, it took 12 and 20 s. The MDR strains representing the ESKAPE-E group showed different behavior, with a 100% reduction in CFU occurring after 8 s of exposure to UV-C for *S. aureus* (MRSA) (3.91 log_10_ CFU/mL), *A. baumannii* (PDR) (3.99 log_10_ CFU/mL) and *K. pneumoniae* (KPC+) (3.95 log_10_ CFU/mL) and 99.5% (2.32 log_10_ CFU/mL) for *P. aeruginosa* (XDR), which was completely reduced (3.84 log_10_ CFU/mL) in the following 10 s exposure time. UV-C causes DNA strand breakage in bacteria, fungi, and human cells, which results in the inhibition of DNA replication. The most important factors related to UV-C lethality include the equipment and treatment parameters, the physicochemical characteristics of the medium, and the type of microorganism [62]. The biocidal effect differs between microorganisms because, due to their factors, some are more resistant than others. Sensitivity to UV-C radiation varies according to the cell wall structure, composition, thickness, presence of proteins that absorb UV-C radiation in the nucleic acid structure, and DNA repair capacity of microorganisms [63]. In this sense, ATCC strains showed less resistance to UV-C treatment than MDR strains from the ESKAPE-E group. These multidrug-resistant pathogens are very difficult to eliminate from the hospital environment with traditional disinfectants [64] and required a longer exposure time to the UV-C system to be completely eliminated. When we compared the CFU count in the assay from a liquid medium (microplate) and solid medium (petri dish), we found that bacterial strains were more affected by UV-C in a tangible medium. UV-C radiation is not transmitted in a medium with a constant intensity equivalent to that generated at the source. From the original, an attenuation effect occurs, due to the absorption of the radiation originally emitted in the medium itself. In a liquid medium, this difference is greater and occurs due to the UV-C absorption of the bacterial suspension since the irradiance decreases as the photons pass through the break, based on the Lambert-Beer law [65].

Unlike chemical disinfection methods, UV-C quickly and effectively inactivates microorganisms through a process based on physical amounts of irradiance or dose frequency, expressed in W.m^−2^, which quantifies the power of the radiation flux received per unit area. As well, energy exposure or UV dose, as said in J.m^−2^, is the main scaling parameter during UV irradiation. This parameter is the product of the intensity emitted by one or more UV sources according to the duration of exposure: Dose (J.m^−2^) = Intensity (W.m^−2^) − sec [64]. Generally, the effectiveness of disinfection largely depends on the radiation dose, because the amount of cellular damage caused by irradiation is proportional to the amount of energy absorbed. However, other factors come into play, such as the distance that separates the UV-C source from the contaminated surface, the nature and concentration of microorganisms, and especially the temperature and humidity of the environment. The particularity of the UV-C band is that it has a powerful germicidal effect. The latter is linked to the fact that radiation is absorbed by DNA and RNA, molecules that support replicative and metabolic functions. In addition, UV-C rays induce different types of damage, acting directly on nucleic acids and proteins [66]. However, not all microorganisms have the same sensitivity to irradiation [67]. Our results clearly demonstrated the efficiency of the UV-C equipment used through the logarithmic reductions observed in ATCCs and MDR strains, depending on the binomial time and distance used.

Many bacterial species possess long filamentous structures known as pili or fimbriae extending from their surfaces. Pili play an important role in adhesion to biotic or abiotic surfaces and biofilm because they facilitate contact between surfaces and cells [68,69]. Images obtained using SEM showed the presence of multiple cellular surface pili-like appendages of variable length that were reduced after UV-C exposure. This result supports the hypothesis that UV-C exposure, by reducing the number of pili present in the strain, is able to interfere with the formation of cell aggregates and cell adhesion to the substrate that favors the decontamination of the evaluated surface. As shoe soles are likely to have a much higher microbial load than most other locations, implementing UV-C equipment that effectively decontaminates shoe soles in hospital settings is expected to represent an important addition to different disinfection strategies already in place.

## 5. Conclusions

Sanitech UV-C shoe sole decontaminator equipment from Astech Company effectively killed in vitro a series of ATCCs and MDR/ESKAPE-E bacteria of sanitary interest, commonly found in the hospital environment. These findings provided important evidence for the effectiveness of UV-C disinfection; therefore, further studies should be encouraged to confirm its efficacy as an adjunct to standard cleaning in reducing HAI-related hospital pathogens.

## Figures and Tables

**Figure 1 ijerph-19-16559-f001:**
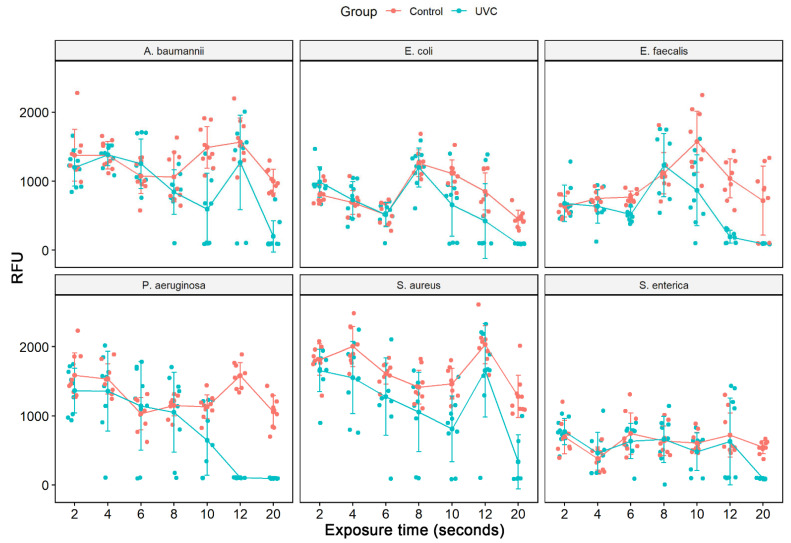
Analysis of cell viability by the addition of resazurin after treatment with UV-C system and control group (no treatment) in different bacterial strains ATCCs (*S. aureus* (ATCC 6538), *P. aeruginosa* (ATCC 15442), *S. enterica* (ATCC 10708), *E. coli* (ATCC 25922), *A. baumannii* (ATCC 19606) and *E. faecalis* (ATCC 29212)). The measured fluorescence intensity (Relative Fluorescence Units, RFU) after the conversion of resazurin to resofurin by viable bacteria was performed either in the control group (no treatment) as in the bacterial suspensions (1:100 dilution) after exposure to the UV-C (2, 4, 6, 8, 10, 12 and 20 s). Results represent values from 3 experiments in triplicate. [times (sec) of UV-C exposure].

**Figure 2 ijerph-19-16559-f002:**
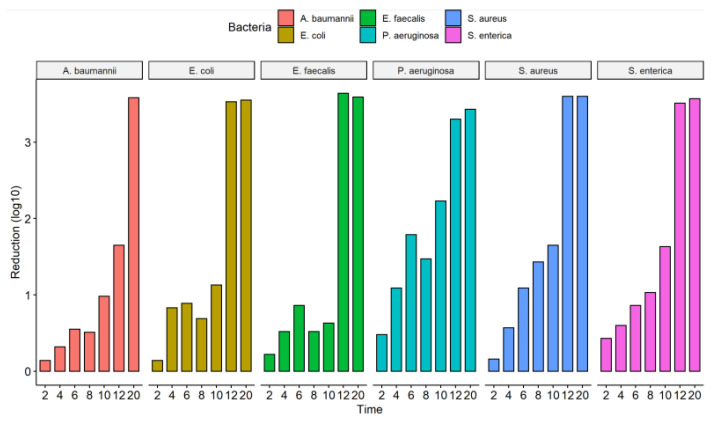
Efficacy of the UV-C system in reducing the number of colony-forming units of the ATCCs strains used and expressed in log_10_ reduction of CFU/mL; CFU, colony forming units.

**Figure 3 ijerph-19-16559-f003:**
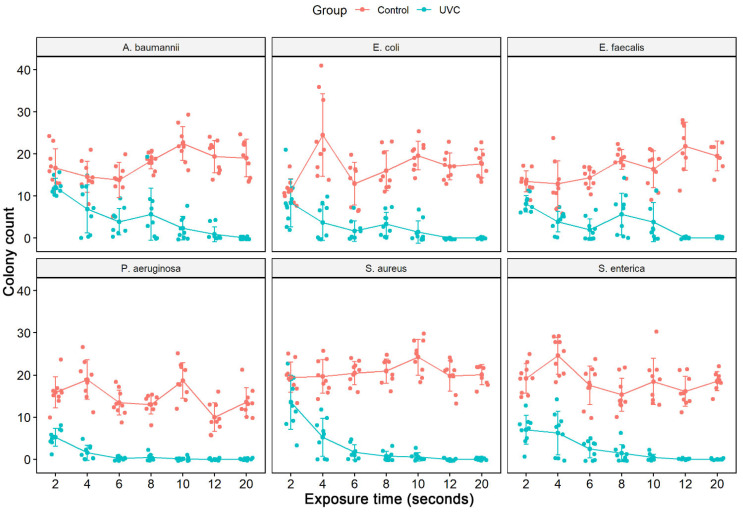
Counting the number of colonies forming units (CFU/mL) of the assay from the liquid medium in different ATCCs bacterial strains (*S. aureus* (ATCC 6538), *P. aeruginosa* (ATCC 15442), *S. enterica* (ATCC 10708), *E. coli* (ATCC 25922), *A. baumannii* (ATCC 19606) and *E. faecalis* (ATCC 29212)). The number of CFU/mL was quantified in the control group (no treatment) and in the bacterial suspensions (1:100 dilution) after exposure to the UV-C system for 2, 4, 6, 8, 10, 12, and 20 s.

**Figure 4 ijerph-19-16559-f004:**
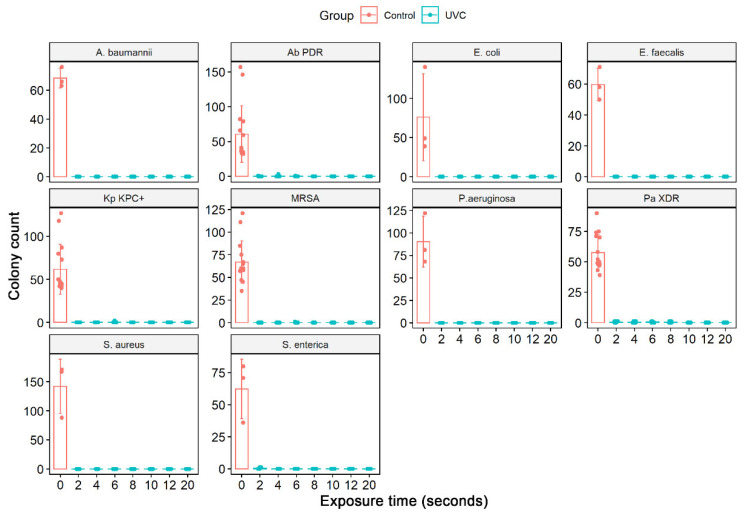
Quantifying the number of CFU/mL in the solid medium of the different bacterial strains ATCCs (*S. aureus* (ATCC 6538), *P. aeruginosa* (ATCC 15442), *S. enterica* (ATCC 10708), *E. coli* (ATCC 25922), *A baumannii* (ATCC 19606), and *E. faecalis* (ATCC 29212) and MDR strains representative of the ESKAPE-E group *S. aureus* (MRSA), *P. aeruginosa* (XDR), *A. baumannii* (PDR) and *K. pneumoniae* (KPC+)). In addition, the number of CFU/mL was quantified in the bacterial suspensions (10^4^ CFU/mL) after exposure to the UV-C system for 2, 4, 6, 8, and 10 s and in the control group (no treatment). Boxes represent median and lines standard deviations.

**Figure 5 ijerph-19-16559-f005:**
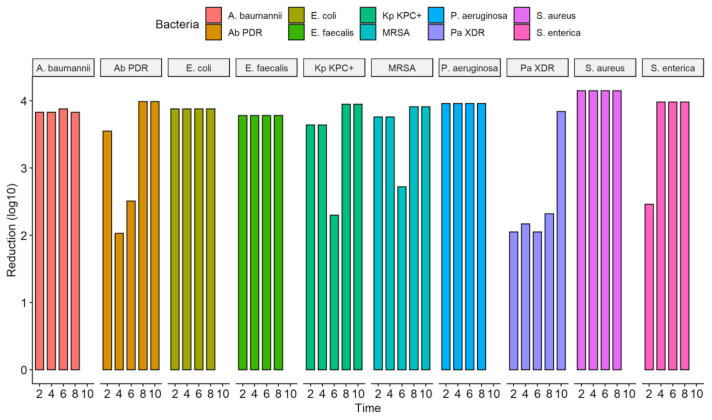
Efficacy of the UV-C system in reducing several bacterial strains expressed in log_10_ reduction of CFU/mL; CFU, colony forming units; *A. baumannii*, *Acinetobacter baumanni*; Ab PDR, *Acinetobacter baumannii* Pandrug-resistant; *E. coli*, *Escherichia coli*; *E. faecalis*, *Enterococcus faecalis*; Kp KPC+, *Klebsiella pneumoniae* produtora de carbapenemase (KPC+); MRSA, *Staphylococcus aureus* resistente à meticilina; *P. aeruginosa*, *Pseudomonas aeruginosa*; *P. aeruginosa* XDR, *Pseudomonas aeruginosa,* Extensively drug-resistant; *S. aureus*, *Staphylococcus aureus*; *S. enterica*, *Salmonella enterica*.

**Figure 6 ijerph-19-16559-f006:**
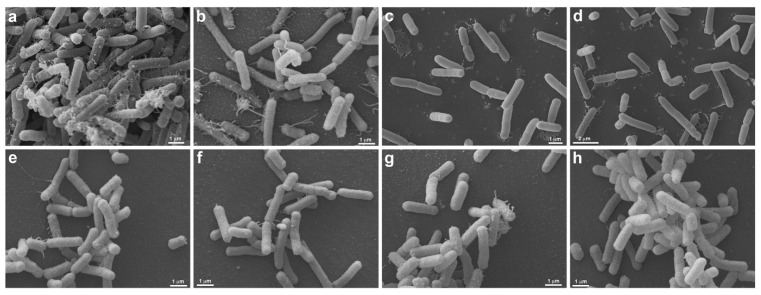
Morphological analysis of UV-C treatment (2 s) by Scanning Electron Microscopy. *S. enterica* (ATCC 10708) (**a**–**d**) and *E. coli* (ATCC 25922) (**e**–**h**) are seen without (**a**,**b**,**e**,**f**) and under UV-C treatment (**c**,**d**,**g**,**h**). Untreated *S. enterica* (**a**,**b**) are elongated, presenting many pili-like appendages emerging from different regions of the cell body. Note that the cells form large aggregates, where more pili-like appendages are seen contacting each other (**a**). After treatment with UV-C, the *S. enterica* (**c**,**d**) still presents the same morphology, but there is a decrease in pili-like appendages and the formation of cell aggregates (**c**,**d**). Both, untreated (**e**,**f**) and treated (**g**,**h**) *E. coli* present the same morphology, the cells are elongated, and less pili-like appendages are observed (**e**–**h**). There is no difference between untreated and treated *E. coli*.

**Figure 7 ijerph-19-16559-f007:**
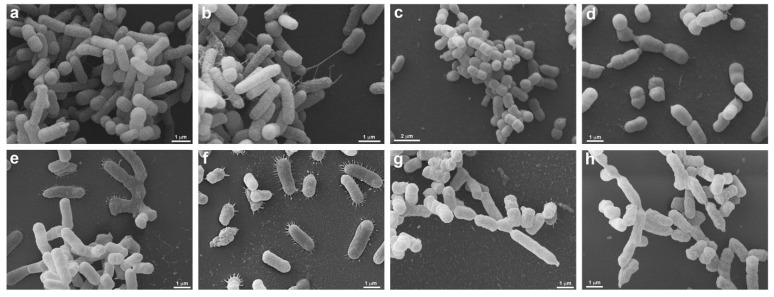
Morphological analysis of *A*. *baumannii* (PDR) and *K. pneumoniae* (KPC+) by Scanning Electron Microscopy. *A*. *baumannii* (PDR) (**a**–**d**) and *K. pneumoniae* (KPC+) (**e**–**h**) are seen without (**a**,**b**,**e**,**f**) and under UV-C treatment by 2 s (**c**,**d**,**g**,**h**). Untreated *A. baumannii* (PDR) (**a**,**b**) presents a heterogeneous morphology, note the presence of cells with different sizes (**a**,**b**), some cells present pili-like appendages extending to other bacteria (**b**). After *A. baumannii* (PDR) treatment (**c**,**d**), the cell population is homogenously smaller (**c**) when compared with untreated ones (**a**), and no pili-like appendages are seen (**d**). UV-C unexposed K. pneumoniae (KPC+) (**e**,**f**) present several pili-like appendages around the elongated cell body (**e**,**f**), these projections are seen mainly when the bacteria are adhered to the substrate (**f**). In treated K. pneumoniae (KPC+) (**g**,**h**) it is possible to note that the cell surface is wrinkled (**g**,**h**), there are different cell sizes (**g**) and there are no pili-like appendages (**g**,**h**).

**Table 1 ijerph-19-16559-t001:** Analysis of cell viability by the addition of resazurin after treatment with UV-C system (10, 12, and 20 s) and control group (no treatment) in different bacterial strains ATCCs (dilution 1:100) *S. aureus* (ATCC 6538), *P. aeruginosa* (ATCC 15442), *S. enterica* (ATCC 10708), *E. coli* (ATCC 25922), *A. baumannii* (ATCC 19606) and *E. faecalis* (ATCC 29212).

Bacterial Strains	Times of Exposure to UV-C *					
	UV-C (10″)	C (10″)	UV-C (12″)	C (12″)	UV-C (20″)	C (20″)
	RFU/% Reduction	Average	RFU/% Reduction	Average	RFU/% Reduction	Average
*S. aureus*	730.6/50.2 *	1465.8	1852.5/8.8	2030.6	334.1/71.9 *	1190.4
*P. aeruginosa*	647.1/42.9 *	1132.8	104.8/93.4 *	1584.3	96.5 /90.9 *	1059.7
*S. enterica*	482.3/11.6	545.3	630.1/3.3	651.3	93.2 /82.9	547.2
*E. coli*	654.8/41.2 *	1112.7	379.2/50.5 *	766.2	94.5 /79.2 *	453.2
*A. baumannii*	595.1/60.1 *	1489.7	1273.2/18.7	1565.6	132.1 /85.3 *	900.52
*E. faecalis*	867.3/44.9 *	1573.6	175.3/83.2 *	1040.9	94.9/85.3 *	646.7

* Statistically significant; **″**, seconds; UV-C, ultraviolet C; C, control; RFU, Relative Fluorescence Units.

**Table 2 ijerph-19-16559-t002:** Reduction in CFU/mL counts after exposure to the UV-C system in ATCCs bacterial strains (*S. aureus* (ATCC 6538), *P. aeruginosa* (ATCC 15442), *S. enterica* (ATCC 10708), *E. coli* (ATCC 25922), *A. baumannii* (ATCC 19606) and *E. faecalis* (ATCC 29212)). The number of CFU/mL was quantified in bacterial suspensions of ATCCs strains (1:100 dilution) after exposure to the UV-C system for 2, 4, 6, 8, 10, 12, and 20 s and in the control group (no treatment).

Bacterial (Strains)	UV-C System Exposure Times													
	2″	C	4″	C	6″	C	8″	C	10″	C	12″	C	20″	C
	CFU/mL Count/% Red/CFU/mL Count		CFU/mL Count/% Red/CFU/mL Count		CFU/mL Count/% Red/CFU/mL Count		CFU/mL Count/% Red/CFU/mL Count		CFU/mL Count/% Red/CFU/mL Count		CFU/mL Count/% Red/CFU/mL Count		CFU/mL Count/% Red/CFU/mL Count	
*S. aureus*	2.710/30.3	3.888	1.044/73.4	3.932	332/9.9	4.088	154/96,3	4.176	110/97.7	4.844	0/100	3.954	0/100	4.022
*P. eruginosa*	1.044/67.1	3.176	310/91.8	3.776	44/98.4	2.688	88/96.6	2600	22/99.4	3.732	0/100	2.000	0/100	2.710
*S. enterica*	1.400/63.6	3.844	1.244/74.7	4.910	488/86.1	3.510	288/90.6	3.066	88/97.6	3.688	0/100	3.222	0/100	3.710
*E. coli*	1.666/26.5	2.266	732/85	4.888	332/87.1	2.576	666/79.2	3.200	288/92.6	3.910	0/100	3.400	0/100	3.532
*A. baumannii*	2.422/27.3	3.332	1.376/52.7	2.910	774/71.9	2.754	1132/69.1	3.666	466/89.6	4.488	88/97.7	3.866	0/100	3.800
*E. faecalis*	1.622/39.7	2.688	776/69.9	2.576	400/86.0	2.866	1132/69.7	3.732	754/76.9	3.266	0/100	4.354	0/100	3.900

CFU count: colony forming units count; red: Reduction; C: control.

**Table 3 ijerph-19-16559-t003:** Reduction in CFU/mL counts in solid media after exposure to the UV-C system in ATCCs and MDR/ESKAPE-E bacterial strains. The quantification of the number of CFUs was performed in bacterial suspensions of ATCCs strains (10^4^ CFU/mL) and MDR strains representative of the ESKAPE-E group (10^4^ CFU/mL) after exposure to the UV-C for 2, 4, 6, 8, and 10 s and control group (no treatment).

Bacterial Strains	UV-C System Exposure Times									
	2″	C	4″	C	6″	C	8″	C	10″	C
	CFU/mL Count/% Reduction/ CFU/mL Count		CFU/mL Count/% Reduction/ CFU/mL Count		CFU/mL Count/% Reduction/ CFU/mL Count		CFU/mL Count/% Reduction/ CFU/mL Count		CFU/mL Count/% Reduction/ CFU/mL Count	
*S. aureus*	0/100	14.200	0/100	14.200	0/100	14.200	0/100	14.200	NP	NP
*P. aeruginosa*	0/100	9.033	0/100	9.033	0/100	9.033	0/100	9.033	NP	NP
*S. enterica*	33/99.7	9.567	0/100	9.567	0/100	9.567	0/100	9.567	NP	NP
*E. coli*	0/100	7600	0/100	7.600	0/100	7.600	0/100	7.600	NP	NP
*A. baumannii*	0/100	6.833	0/100	6.833	0/100	6.833	0/100	6.833	NP	NP
*E. faecalis*	0/100	5967	0/100	5.967	0/100	5.967	0/100	5.967	NP	NP
*S. aureus* (MRSA)	0/100	5.711	0/100	5.711	11/99.8	5.711	0/100	8.183	0/100	8.183
*P. aeruginosa* (XDR)	44/99.1	4.911	33/99.3	4.911	44/ 99.1	4.911	33/99.5	6.983	0/100	6.983
*A. baumannii* (PDR)	0/100	3.556	33/99.1	3.556	11/ 99.7	3.556	0/100	9.817	0/100	9.817
*K. pneumoniae* KPC+	0/100	4.356	0/100	4.356	22/ 99.5	4.356	0/100	8.917	0/100	8.917

Caption: Cont. CFU/mL: Count of colony forming units/mililitre; C: Control; NP: Not performed.

## Data Availability

Not applicable.

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
