# Peer review of "Effectiveness Evaluation of a UV-C-Photoinactivator against Selected ESKAPE-E Pathogens"

_ijerph, 2022, doi:10.3390/ijerph192416559_

Round 1
Reviewer 1 Report
Revision of manuscript ijerph-2047262
Effectiveness Evaluation of a UVC-Photoinactivator against Selected ESKAPE-E Pathogens
by Rangel et al.
General comments
The manuscript reports the inhibitory effect of a UV-C photoinactivator against some important pathogens.
The topic is very important and worthy of investigation. However, the manuscript requires some revisions.
Specific comments
UVC should be changed in UV-C in the entire manuscript.
Introduction
l. 51-52: use italics for bacterial species names. See also l. 305 and l. 470.
l. 53: change “families” with “species and genus”
l.134-136: change UVA with UV-A. The same for UVB and UVC.
Materials and methods
l. 154: “Standard strains” should be deleted
l.155: choleraesuis should be written not italics and with capital letter.
l. 157: “n°” should be deleted in “ATCC nº”. See also l. 305.
l. 161 and 163: definitions for acronyms PDR and XDR should be added. Please also add antibiotic names towards which the PDR and XDR bacteria showed resistance.
l. 175: please indicate the total UV-C intensity (microwatts per square centimeter) of the equipment.
l. 179: change Tryptona in Tryptone
l. 190: since bacterial suspensions were made in PBS, why LB broth has been used as negative control?
l. 209: what is CLB?
l. 233: it seems to me that something is missing in the formula. Please check.
Results
Table 1: Is fluor for “RFU”? If so, please change fluor with RFU. Moreover, after “ATCC 19606” a parenthesis is missing.
l. 280: paragraph 3.1.2. Quantification of Colonies is redundant with Table 2 and should be strongly synthesized.
l. 318: change ATTC with ATCC and Ml with ml.
l. 324: Change Tabela with Table.
Discussion
Discussion section should be strongly synthesized, in particular omitting redundancies with Results and avoiding to repeat inactivation percentages.
l. 409: Change n with In
l. 424: resazurin is a well- known reagent, and the detailed explanation of its mechanism should be omitted. Please delete lines 424 to 437.
Results should be discussed taking into account the differences in the UV doses needed to inactivate bacterial species used in this study.
A comment about the different impact between UV-C lamps and LEDs should be added.

Author Response
Reviewer 1:
General comments: The manuscript reports the inhibitory effect of UV-C photoinactivation against some important pathogens. The topic is very important and worthy of investigation. However, the manuscript requires some revisions.
R: Thank you
Specific comments
UVC should be changed in UV-C in the entire manuscript.
R: UVC was changed in UV-C in the entire manuscript.
Introduction
- 51-52: use italics for bacterial species names. See also l. 305 and l. 470.
R: All were corrected.
- 53: change "families" with "species and genus."
R: It was changed.
l.134-136: change UVA with UV-A. The same is true for UVB and UVC.
R: All were changed.
Materials and methods
- 154: "Standard strains" should be deleted.
R: It was deleted.
l.155: choleraesuis should be written not italics and with a capital letter.
It was corrected.
R: l. 157: "n°" should be deleted in "ATCC nº." See also l. 305.
Both were deleted.
- 161 and 163: definitions for acronyms PDR and XDR should be added. Please also add antibiotic names towards which the PDR and XDR bacteria showed resistance.
R: The definitions for acronyms PDR and XDR were added.
XDR was defined as "nonsusceptibility to at least one agent in all but two or fewer antimicrobial categories" (i.e., bacterial isolates remain susceptible to only one or two categories), and PDR was defined as "nonsusceptibility to all agents in all antimicrobial categories" (Y, Falagas ME, Giske CG, et al. Multidrug-resistant, extensively drug-resistant and pan drug-resistant bacteria: an international expert proposal for interim standard definitions for acquired resistance. Clin Microbiol Infect 2012; 18 (3): 268-81.).
- 175: please indicate the equipment's total UV-C intensity (microwatts per square centimeter).
R: The intensity was 15.150 µW/cm2.
- 179: change Tryptona in Tryptone
R: It was changed.
- 190: since bacterial suspensions were made in PBS, why has LB broth been used as a negative control?
R: Because in the Cell Viability experiment, cultures were diluted in LB broth after exposure of microplates containing bacterial strains to the UV-C system.
- 209: what is CLB?
R: CLB meant LB broth and was corrected in lines 221 and 227.
- 233: something is missing in the formula. Please check.
R: The formula was corrected.
Results
Table 1: Is fluor for "RFU"? If so, please change fluor with RFU.
R: Yes. It was changed.
Moreover, after "ATCC 19606," a parenthesis is missing.
R: The ATCC number was taken from all strains in the table (another reviewer's request).
- 280: paragraph 3.1.2. Quantification of Colonies is redundant with Table 2 and should be strongly synthesized.
The text was synthesized.
- 318: change ATTC with ATCC and Ml with ml.
Both were changed.
- 324: Change Tabela with Table.
It was changed.
Discussion
The discussion section should be strongly synthesized, in particular, omitting redundancies with Results and avoiding to repeat inactivation percentages.
- 409: Change n with In
It was corrected.
- 424: resazurin is a well-known reagent, and the detailed explanation of its mechanism should be omitted. Please delete lines 424 to 437.
Lines were deleted.
Results should be discussed, considering the differences in the UV doses needed to inactivate bacterial species used in this study.
A comment about the different impacts between UV-C lamps and LEDs should be added.
It was added in lines 144-148.

Reviewer 2 Report
The aim of the article entitled: Effectiveness Evaluation of a UVC-Photoinactivator against Selected ESKAPE-E Pathogens was to evaluation if the SANITECH UVC shoe sole decontaminator equipment can be able to reduce this migration risk effectively in different pathogens.
The article seems interesting for a certain group of recipients, and new devices should be tested for the effectiveness of pollutant migration, which is a great advantage of the conducted research. However, there are some considerations and considerations that need to be clarified in order to better understand and communicate the results.
50-51: Strain names should be in italics
63: Correct the citation - in accordance with the guidelines of the journal, e.g. add the year
96: Please add the short name of the strain in brackets and then use only the abbreviation.
160 and 161 Abbreviations are not explained
169 Degree symbol needs improvement - superscript
187 Nowhere in the article is it explained why the authors used such exposure times. Any literature references?
189/ 190 Preparation of the Negative Control, why PBS and LB were used and not PBS and broth in which the sharks were grown
207 Why 5uL? In the surface culture, 100 µl are poured out, and then the appropriate calculations are made.
231 I have a big problem with reading the results. The authors give and often refer to CFU only without reference to volume. The correct unit is CFU/mL. Please fill in the gaps in the text.
231 Why p=0.01 and not 0.05? 235 What does B mean in the formula? B- refers to CFU/ml without irradiation.
240 There are no companies from which the reagents were taken 243 100%? I suppose it was 99.9 or 99.6%
249-250 This sentence is grammatically unclear and needs improvement
Tab 1 - no strain reference numbers are needed, only names. In addition, the captions under the table are hardly legible
Fig 1. No legend Table 2-Cfu for what volume? Was it converted to 1 ml or is it a value in 5 uL??
Fig 2 as above
325 and 326 make up units
Fig. 4 Results are shown in boxes. What do they mean? What range? Is it the standard deviation or the median?
Table 3 why can't there be one unit? There is no need to specify the strain numbers since it has been presented before
369 pellets? Please expand on what makes pellets. Could it be the EPS produced by the strains?
379 SEM analysis - descriptions are confusing and require explanation
409 typo
412 unit
Author Response
Reviewer 2:
The aim of the article entitled: Effectiveness Evaluation of a UVC-Photoinactivator against Selected ESKAPE-E Pathogens was to evaluation if the SANITECH UVC shoe sole decontaminator equipment can be able to reduce this migration risk effectively in different pathogens.The article seems interesting for a certain group of recipients, and new devices should be tested for the effectiveness of pollutant migration, which is a great advantage of the conducted research. However, there are some considerations that need to be clarified in order to better understand and communicate the results.
50-51: Strain names should be in italics.
R: The names were corrected.
63: Correct the citation - in accordance with the guidelines of the journal, e.g., add the year
R: It was corrected.
96: Please add the short name of the strain in brackets and then use only the abbreviation.
R: The strain name appears in the text for the first time, so it was not abbreviated
160 and 161 Abbreviations are not explained
R: We added the explanation of abbreviations in lines 161 to 164.
169 Degree symbol needs improvement – superscript
R; It was superscript.
187 Nowhere in the article is it explained why the authors used such exposure times. Any literature references?
R: There is no reference in the literature. The authors decided to use exposure times ranging from 2 seconds to 12 seconds, as it would be a reasonable time for the individual to wait on the equipment. Since the maximum decontaminator timer time was 20 seconds, we also tested this time. This explanation was added in lines 188 to 191.
189/ 190 Preparation of the Negative Control, why PBS and LB were used and not PBS and broth in which the sharks were grown
R: Only PBS was used as the negative control. It was corrected (lines 194-195).
207 Why 5uL? In the surface culture, 100 µl are poured out, and the appropriate calculations are made.
R: We used 5uL to facilitate the colony count.
231 I need help with reading the results. The authors give and often refer to CFU only without reference to volume. The correct unit is CFU/mL. Please fill in the gaps in the text.
R: When the authors performed the cell viability experiment and then colony counting, the bacterial suspensions in PBS initially at a concentration of 106 CFU/mL) were exposed to UV-C. Afterward, 1:10, 1:100, and 1:1000 dilutions were made. As the initial culture told had its concentration changed after the dilutions above, we cannot say that these concentrations became 105, 104, and 103 CFU/mL; therefore, we refer only to CFU without the volume reference.
231 Why p=0.01 and not 0.05?
R: The correct is 0.05.
235 What does B mean in the formula? B- refers to CFU/ml without irradiation.
R; The formula was corrected.
240 There are no companies from which the reagents were taken 243 100%? I suppose it was 99.9 or 99.6%
R: Both were corrected.
249-250 This sentence is grammatically unclear and needs improvement
R: The sentence was corrected to "Cell viability analysis in solution was evaluated by spectrophotometric reading of the measurement of resazurin to resorufin reduction."
Tab 1 - no strain reference numbers are needed, only names. In addition, the captions under the table are hardly legible.
R; It was corrected.
Fig 1. No legend
R: It was corrected.
Table 2-Cfu for what volume? Was it converted to 1 ml, or is it a value in 5 uL??
R: Cfu was converted to 1mL. It was corrected.
Fig 2 as above
R: It was corrected.
325 and 326 make up units
It was corrected.
Fig. 4 Results are shown in boxes. What do they mean? What range? Is it the standard deviation or the median?
The boxes represent the median and line the standard deviation.
Table 3 why can't there be one unit? There is no need to specify the strain numbers since it has been presented before
It was corrected.
369 pellets? Please expand on what makes pellets. Could it be the EPS produced by the strains?
Precipitate (“pellet”). It was explained in line 370.
379 SEM analysis - descriptions are confusing and require explanation
It was altered.
409 typo
It was corrected.
412 unit
It was corrected.

Round 2
Reviewer 2 Report
The authors reply is acceptable and clarifies the questionable points